# What Ails One-Shot Image Segmentation: A Data Perspective

**Abhinav Patel**[*]
Adobe Inc.
abpatel@adobe.com

**Anirudha Ramesh**[*]
IIIT Hyderabad, India
anramesh@iiit.ac.in

**Tejas Shimpi**[*]
IIT BHU, India
tshimpi@iitbhu.ac.in

**Mayur Hemani**[*]
Adobe Inc.
mayur@adobe.com

**Balaji Krishnamurthy**
Adobe Inc.
kbalaji@adobe.com

## Abstract

One-shot image segmentation (OSS) methods enable semantic labeling of image pixels without supervised training using an extensive dataset. They require just one example (image, mask) pair per target class. Most neural-network based OSS methods train on a large subset of dataset classes, and are evaluated on a disjoint subset of classes. We posit that the data used for training induces negative biases and affects the accuracy of these methods. Specifically, we present evidence for a *Class Negative Bias* (CNB) arising from treating non-target objects as background during training, and *Salience Bias* (SB), affecting the segmentation accuracy for non-salient target class pixels. We demonstrate that by eliminating CNB and SB, significant gains can be made over existing state-of-the-art. Next, we argue that there is a significant disparity between real-world expectations from an OSS method and its accuracy reported on existing benchmarks. To this end, we propose a new evaluation dataset - Tiered One-shot Segmentation (TOSS) - based on the PASCAL $5^i$ and FSS-1000 datasets, and associated metrics for each tier. The dataset enforces consistent accuracy measurement for existing methods, and affords fine-grained insights into the applicability of a method to real applications. The paper includes extensive experiments with the TOSS dataset on several existing OSS methods. The intended impact of this work is to point to biases in training and introduce nuances and uniformity in reporting results for the OSS problem. The evaluation splits of the TOSS dataset and instructions for use are available at https://github.com/fewshotseg/toss.

## 1 Introduction

Semantic image segmentation assigns class labels to pixels in an image. It is useful for diverse applications such as image editing ([1]), content-based retrieval ([32, 14]), medicine ([11, 9]) and art ([10]). Accurate segmentation on a small number of classes can be achieved by training deep neural networks on large datasets. However, when the number of classes is large, or the training examples are hard to obtain, a supervised approach cannot be used. Semantic segmentation of images with very few training examples is referred to as the *Few-shot Segmentation* (FSS) [26] problem. When exactly one example is available, it is referred to as *one-shot* image segmentation (OSS).

Recent OSS methods (like [37, 22, 19, 33, 39, 18, 36, 28, 12, 21]) predict a binary mask for each class of interest. They employ a dual-branched neural network (see Figure 1), where one branch is

---

[*]equal contribution

35th Conference on Neural Information Processing Systems (NeurIPS 2021) Track on Datasets and Benchmarks.

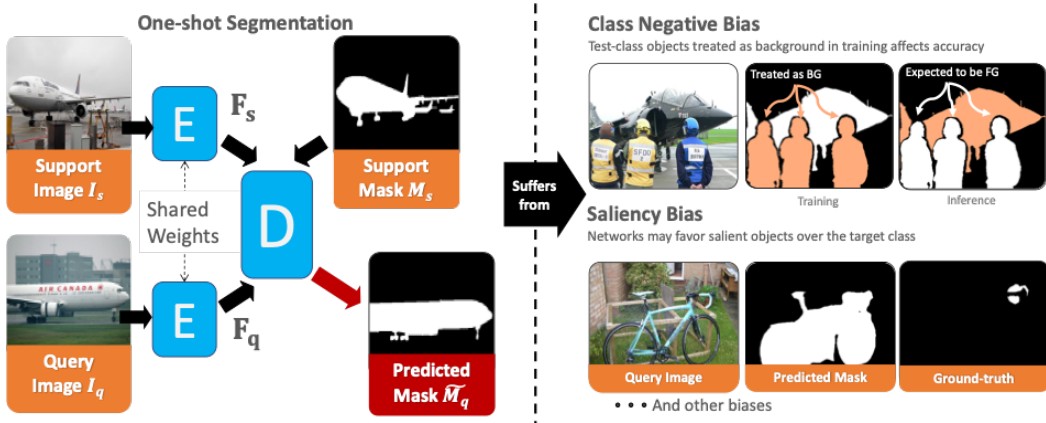

Figure 1: The one-shot segmentation task. A support image $I_s$ and a query image $I_q$ are both passed through a common encoder $F$. The support features $F_s$, the support mask $M_s$ and the query features $F_q$ are fused and decoded to predict the query mask $M_q$. In this work, we show negative inductive biases arising from treating unknown objects as background during training (Class Negative Bias), and from favoring salient objects over the target class (Salience Bias).

supplied a set of *support* images and binary masks corresponding to the target class as input, while the other branch is given a *query* image. The support branch produces a target-class representation which is fused with the query-branch features, and decoded to predict the binary mask. Most solutions train for OSS and extend the inference apparatus for FSS. While the accuracy of OSS solutions has improved considerably since the problem was introduced in [26], the best solutions still fall short of the accuracy of fully supervised methods. We posit that these networks have certain negative inductive biases caused by the improper use of the training datasets. These biases manifest as very low quality segmentation masks for certain kinds of input images. We also observe the need for an evaluation approach that offers fine-grained insights into the problem. It would afford better alignment with real-world applications by exposing the actual limitations in an incumbent solution. For example, an application that requires extraction of relatively isolated objects may work well with an OSS method that has a high accuracy for salient objects. The method may be chosen over another with a better average accuracy over a diverse set of benchmark images. In this work, we discuss negative inductive biases in existing methods and propose an evaluation dataset for cogent reporting.

The OSS training protocol described in [26] allows treating foreground objects that are not in the training set as background pixels. We posit that this introduces a negative inductive bias (termed *Class Negative Bias* (CNB) in OSS networks. To corroborate this position, we experiment with multiple baseline methods ([7, 28, 33]) on the PASCAL $5^i$ dataset (see Section 3.2). We find a correlation between the drop in OSS test accuracy of certain classes with the presence of objects of these classes in training images where they are treated as background (2). We introduce a salience-guided training set selection strategy (SGTSS, section 3.2.1) to mitigate CNB. It selects only those images for training where the target class pixels are in the salient regions. This weakens the effect of any distracting objects present in the training set images. We present evaluation on the PASCAL-$5^i$ dataset.

Next, we notice that for images where the target-class pixels are in visually salient regions, the OSS accuracy is significantly better. Also, many existing networks ([37, 30, 7, 33]) produce good results even without any support images (see section 3.3.1). Both experiments indicate a strong bias favoring salient objects over non-salient ones. In section 3.4, we elucidate the effect of mitigating this Salience bias (SB).

These biases in existing networks demonstrate that the accuracy of segmentation in the OSS framework varies significantly with the input images. Current benchmark datasets (PASCAL $5^i$, COCO $20^i$, and FSS-1000, [26, 22, 13]) do not account for these variations in input. The mean intersection-over-union (mIoU) score used with these benchmarks also conceal nuances about the shortcomings of the existing methods. We propose a fine-grained evaluation benchmark based on the PASCAL $5^i$ and FSS-1000 datasets. These evaluation splits, termed *Tiered-OSS* (TOSS), consider the inherent difficulty in segmenting an image with a fully-supervised method, the salience of the target regions, and the effect of the support image. Specifically, we introduce three tiers - the *Query Complexity*

tier with varying degrees of complexity of query images, the *Support Cognizance* tier with degrees of support-query similarities, and the *Generalization* tier to gauge generalization to unseen classes. Corresponding to these tiers, we propose scores based on the prevalent mIoU metric that indicate the accuracy for each task. These nuances in reporting also improve alignment with the real-world expectations for an OSS solution. To summarize, this paper contributes the following:

- A detailed analysis of two inductive biases that negatively affect the accuracy of one-shot segmentation methods - *Class negative bias* from treating non-target object classes as background in training, and *Salience bias* - favoring salient objects to target class.
    - Documentation of the effect of mitigating these biases using Salience-guided training set selection (SGTSS) for reducing class negative bias, and selective salience suppression at inference for salience bias.
- Tiered-OSS (TOSS) - a fine-grained, tiered one-shot evaluation dataset based on the PASCAL $5^i$ and FSS-1000 datasets, and the corresponding scores for more nuanced reporting of OSS accuracy, and aiding discovery and mitigation of biases in existing methods.

## 2   Related Work

The present work is related to existing few-shot segmentation (FSS) research, as well as to existing benchmark datasets and metrics for OSS.

**Few-shot Segmentation (FSS) Methods:** Shaban et al. introduced a two-branched deep neural-network based framework in [26] for semantic image segmentation using a single example. The method extracts features from the support image (and mask) to predict the weights of the pixel classifier layer of the query branch. This work was superseded by Rakelly *et al.* [25] with the introduction of *late fusion* where the support mask is fused after the feature extraction step. This approach is the template of much of the later work. The variations on it include the use of prototypical learning [5, 30, 19, 33, 17, 12], the use of dense matching [38, 22, 37, 7, 39, 21, 35], and the use of graph-based attention mechanisms [36, 19, 29]. Many methods used fixed backbone networks for feature extraction (like [37, 7, 39]), while others allow for training of all features ([38, 18]) to learn co-occurring features between the support and query. Intermediate class representation is another factor - Zhang *et al.* [38] employ a masked average pooling (MAP) operation on the support features and mask, [22] uses relevance weighting on top of that, and [33] replaces MAP with mixture models [33]. We are concerned with extracting biases that affect accuracy for OSS. In [2], Azad *et al.* consider the bias of convolutional layers towards high-frequency information in the input images, and propose how to mitigate it. We focus on data-induced biases and demonstrate that most of these methods still suffer from them.

**OSS Datasets and Metrics:** The PASCAL $5^i$ dataset proposed in [26] is based on images and mask annotations from the PASCAL VOC dataset [6] and the Semantic boundaries dataset [8]. It consists of 12031 images, with pixel-level annotations for 21 classes (class 0 is background). The annotations are split into 4 folds, each with 5 test classes, and 15 training classes. To address the small size of the PASCAL $5^i$ dataset, Nguyen *et al.* [22] propose the COCO $20^i$ dataset based on the COCO segmentation dataset [15]. This too comprises of 4-folds, but each fold has 60 training classes, and 20 test classes. The best test accuracy for COCO $20^i$ is significantly lower than that for PASCAL $5^i$. The FSS-1000 dataset introduced by Li *et al.* in [13] further scales the number of classes to 1000, with 760 training classes and 240 test classes, with 10 image-mask pairs per class. These datasets suffer from significant variance in results because the exact pairs of test images are not specified. Another recent dataset is the LVIS-OneShot [20] but its evaluation is not included in our work due to its unavailability.

Most one-shot segmentation methods report the mean over test classes of the intersection-over-union (mIoU) of the predicted mask and the ground-truth mask for each fold of the dataset. Some papers also report foreground-background IoU, disregarding the classes though it does not reflect OSS accuracy adequately. Computing mIoU over the folds conceals the behavior of the networks for different kinds of input. We introduce 4 scores defined over 3 tiers of test data to indicate the accuracy considering query complexity, precision, and generalizing to unseen classes.

In the next section, we discuss biases introduced into OSS networks by the training data used. In section 4 we describe our proposed dataset for improved evaluation of OSS.

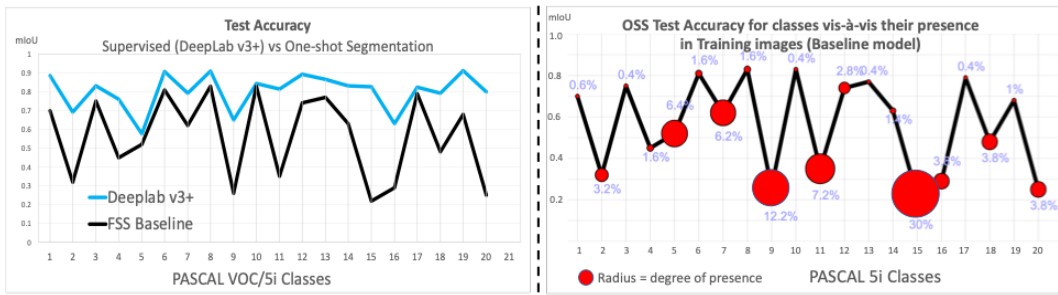

Figure 2: Test accuracy of the baseline network with different PASCAL $5^i$ classes. The left column compares the test accuracy of Deeplab v3+ [4] network on the PASCAL $5^i$ classes (same as PASCAL VOC) with the 1-shot accuracy of the baseline network. The right column indicates the fraction of training set images containing objects of a particular class as a non-target class. For classes that have a conspicuous presence as distraction in the one-shot training sets, severe drops in accuracy (from supervised to one-shot) occur. This indicates class-negative bias in the network.

## 3 Data-induced Negative Inductive Biases

Negative inductive biases in a trained OSS neural-network can cause it to have skewed accuracy over different kinds of input images. We discuss the evidence for the existence of two such data-induced biases, and their mitigation. The evidence is collected using a baseline network derived from the dense-matching framework of CANet [37] with the dual-prediction and background-attention ideas from SimPropNet [7]. The network is trained and evaluated on the PASCAL $5^i$ dataset.

### 3.1 Overview of the One-shot Segmentation Task

The one-shot segmentation problem is formulated as follows: Given a single *support* image-mask $(I_s, M_s)$ pair as an example of the semantic segmentation of a target class $C$, predict the binary segmentation mask $M_q$ of objects of that class in a different *query* image ($I_q$). As described in [26], OSS networks are trained with (support, query) pairs chosen at random from the training set of a dataset fold. They are tested on pairs from the test set of the fold. The PASCAL $5^i$ dataset has 20 classes, which are split into 4 folds, each with 5 test-classes and 15 training classes. We train our baseline network on the PASCAL $5^i$ dataset and conduct multiple experiments to identify its biases.

### 3.2 Class Negative Bias

We define *Class Negative bias* (CNB) as the inductive bias in OSS networks that results from treating objects not present in the training classes as background during training, and causes error in segmenting such objects during test. We first present evidence for this bias and then propose a mitigation strategy for it.

**Evidence for Class Negative Bias** Our evidence for CNB is based on three observations:

1. An incommensurate drop in test accuracy with baseline OSS network, for classes numbered {2, 4, 9, 11, 15, 16, 18, 20}, compared to the accuracy with a pre-trained DeepLab v3+ [4] (Figure 2).
2. The conspicuous presence of objects of these classes in the corresponding OSS training sets where they are treated as background, and
3. The gain in accuracy in OSS when the network is trained without images containing these test-class objects (see Table 1, *LeaveOut*). For example, the gain in accuracy for the person class increases from 31% to 49% (See supplemental material for more details).

Considered together, the results of these experiments indicate that there a negative bias is introduced due to the presence of test-class objects in training images, and them being treated as background. Next, we discuss CNB mitigation.

| OSS Method | PASCAL $5^i$ mIoU | | | | |
|---|---|---|---|---|---|
| | Fold-1 | Fold-2 | Fold-3 | Fold-4 | Mean |
| Base | 51.06 | 64.35 | 55.70 | 50.44 | 55.39 |
| Base+Leaveout | 51.10 | 66.32 | 58.07 | 52.16 | 56.9 |
| Base+SGTSS | 54.5 | 66.71 | 61.65 | 53.26 | 59.03 |
| RPMM$^{*+}$ | 45.72 | 61.40 | 47.50 | 44.31 | 49.73 |
| RPMM+Leaveout | 53.78 | 64.92 | 55.11 | 47.60 | 55.35 |
| RPMM+SGTSS | 50.3 | 62.46 | 55.13 | 47.81 | 53.93 |
| PFENet$^+$ | 52.87 | 65.088 | 50.16 | 51.90 | 55.01 |
| PFENet+Leaveout | 56.83 | 68.49 | 57.36 | 55.05 | 59.43 |
| PFENet+SGTSS | 55.48 | 65.63 | 58.24 | 49.17 | 57.13 |

Table 1: Evidence and mitigation for Class Negative Bias in multiple existing networks. Leaveout refers to filtering out images that have test-class objects from the training set. SGTSS = Salience guided training set selection. The *leave-out* approach is technically not permitted in a true one-shot segmentation setting. $^*$ RPMM is re-trained without their class-based omission implementation. Also note that the test files used contains 15000 pairs for each fold to reduce statistical error. $^+$ Quoted mIoU numbers are as per our test splits.

### 3.2.1 Effect of alleviating Class Negative Bias

To alleviate the class negative bias is to train the network to treat non-target foreground objects not as background. This could be done by eliminating all images that have non-target class objects from the training set (as in the official implementation for [33]). The disadvantage of this approach is that it assumes knowledge of test-class objects in the images. In a true one-shot setting, this cannot be known. We observe that if the salient regions of an image are occupied by pixels of classes not in the training set, it is likely that the image has *distracting* objects. These images are eliminated from the training set as described below.

**Salience-guided Training Set Selection (SGTSS):**   A salience mask for each training image is computed using a pre-trained $U^2$-Net [23] network. The salience mask is used to compute a salience score for each image as follows:

$$K_i = \frac{\|(\bigcup_{t \in T} M_i^t) \cap S_i\|}{\|S_i\|} \tag{1}$$

Here the $M_i^t$ denotes the masks of all objects that belong to some training class ($t \in T$), and $S_i$ is the salience map for the $i^{th}$ image . The score is the ratio of the intersection area of any training class mask with the salient regions, and the area of the salient region. It is the fraction of the salient regions occupied by training class pixels. $K_i$ is computed for each image and the ones with values lower than the median are eliminated from the training set. Thus the training set comprises of images with mostly training class pixels in their visually salient regions.

Table 1 presents the results with SGTSS. There is a significant gain across folds despite the training set being actually smaller. Next, we discuss the salience bias present in these networks.

## 3.3 Salience Bias

We observe that many encoder-decoder based neural-networks for one-shot image segmentation are most accurate with visually salient regions of an input image.

### 3.3.1 Evidence for Salience Bias

Table 2, column A presents some unexpected results with the baseline network on the PASCAL $5^i$ dataset. Supplying no support mask as input to the baseline network still produces query masks with comparable accuracy. Supplying no support image (and no mask) as input still yields some mask with accuracy comparable to the hard cases with both inputs. Supplying a support input from a different class also yields a reasonable result. These results indicate that the network does not always consider

| A: Effect of Different Support Input Types | | | | | | B: Effect of Salience | | |
|---|---|---|---|---|---|---|---|---|
| **Support Type** | **PASCAL** $5^i$ **mIoU** | | | | | **Method** | **mIoU** | |
| | $5^1$ | $5^2$ | $5^3$ | $5^4$ | Mean | | Salient | Non-Salient |
| Base | 55.2 | 66.3 | 54.6 | 49.5 | 56.4 | Baseline | 61.0 | 38.3 |
| $C(I_q){\neq}C(I_s)$ | 46.9 | 44.2 | 36.3 | 37.8 | 41.3 | RPMM | 57.8 | 33.2 |
| $M_s{=}0$ | 35.4 | 12.9 | 24.1 | 24.8 | 24.3 | PFENet | 61.9 | 38.2 |
| $I_s{=}0$ | 19.1 | 36.7 | 31.0 | 22.4 | 27.3 | | | |
| $C(I_q){\neq}C(I_s){+}M_s{=}0$ | 37.4 | 21.5 | 30.5 | 31.7 | 30.3 | | | |

Table 2: Evidence for Salience bias. Column A shows that the baseline network predicts a query mask despite receiving spurious support input. Column B shows that the accuracy is skewed across different methods in favor of salient objects. $C(I_q)$ and $C(I_s)$ refer to the target class for the query and support images respectively. $5^x$ refers to the folds of the PASCAL-$5^i$.

the signal from the support branch, and is still able to produce some non-trivial results for certain images. Next, we split the test set of each fold by the degree of salience of the target class pixels. The degree of salience is approximated by the overlap (IoU) of the target mask with a salience mask of the query image obtained from a pre-trained $U^2$-Net network [23]. We notice that the mIoU for the salient split is greater than the non-salient split by a large margin (Table 2, column B). This clearly indicates the preference of the networks for salient target pixels, thus confirming the bias.

### 3.4 Effect of mitigating Salience Bias

We consider an oracle case where we have prior knowledge of whether the target pixels are in the salient region or not. This prior can be generated with a human user providing feedback on the predicted segmentation from an OSS network. For the non-salient case, we suppress the salient regions in the input query image at inference time, using a wide-radius Gaussian blur. The salient regions for an image are obtained using a pre-trained $U^2$-Net (trained on the DUTS-TR dataset [31]). Table 3 indicates the significant improvement we obtain in the accuracy by eliminating the Salience bias from the OSS networks.

| OSS Method | PASCAL $5^i$ mIoU | | | | |
|---|---|---|---|---|---|
| | Fold-1 | Fold-2 | Fold-3 | Fold-4 | Mean |
| Base | 51.06 | 64.35 | 55.70 | 50.44 | 55.39 |
| Base+SSSI | 52.99 | 66.23 | 58.04 | 52.65 | 57.48 |
| RPMM* | 45.72 | 61.40 | 47.50 | 44.31 | 49.73 |
| RPMM+SSSI | 46.26 | 62.12 | 48.86 | 45.86 | 50.52 |
| PFENet | 52.86 | 65.09 | 50.16 | 51.90 | 55.01 |
| PFENet+SSSI | 55.13 | 67.30 | 52.97 | 54.40 | 57.44 |

Table 3: Effect of Selective Salience Suppression at Inference (SSSI) with prior knowledge of whether the target pixels are in the salient region or not yields significant gain in accuracy.

## 4 Tiered Evaluation for OSS

We posit that having an evaluation dataset that - i.) maps better to real-world image segmentation expectations, and ii) provides a nuanced view of the accuracy of any OSS method can be a very useful tool in advancing the state-of-the-art for the problem. To this end, we propose a new multi-faceted evaluation dataset - the *Tiered One-shot Segmentation* (TOSS) dataset, and the associated metrics. In this section, we present the structure of the dataset, the corresponding measurements, the results of different OSS methods on the dataset and insights into the problem using the dataset.

### 4.1 Tiered One-shot Segmentation Evaluation

The dataset is based on the PASCAL $5^i$ dataset proposed by Shaban et al. in [26], and the FSS-1000 dataset [13]. We retain the training folds of the PASCAL $5^i$ datasets and focus on the evaluation only.

For each fold, the TOSS dataset comprises of three tiers of measurements, each of which has pairs of query-support images. These tiers are evaluated on the predictions of the incumbent OSS network:

1. **Tier 1: Query Complexity** - to 4 splits of support-query pairs constructed by varying complexity of the query image.
2. **Tier 2: Support Cognizance** - to gauge the influence of the support image on the segmentation outcome.
3. **Tier 3: Generalization** - to evaluate the generalization of the network to images of a different domain. The images in this tier are taken from the FSS-1000 dataset [13].

Together these tiers indicate the accuracy of a network for various kinds of input data. Figure 3 depicts the tiers of the evaluation dataset.

### 4.1.1 Query Complexity (QC)

For each PASCAL $5^i$ fold, we use its test classes to construct our evaluation splits. The test set is partitioned on two axes - the accuracy of a supervised method (DeepLab v3+ [4]) for a class, and the overlap of the class mask with the salience mask for the image. Four splits of (query, support) pairs are then constructed by sampling the query images from corresponding partitions. These four splits are ordered as *{(easy, salient), (hard, salient), (easy, non-salient), (hard, non-salient)}*. To marginalize the effect of the support image, we sample the support images uniformly from all of the partitions. Figure 3 depicts query images from these splits.

**Attribute Choices:** The accuracy of segmentation depends upon several factors including scene illumination, materials and occlusion. We consolidate all of these into a single axis for a given image - the IoU value for the target class with a supervised method. For the second axis of partitioning, we use the overlap of the target pixels with the salient regions. This attribute has high mutual information with respect to the IoU of the predicted mask. It is possible to consider other factors (object size, offset from center, etc.). However, to keep the number of test splits manageable, we only partition with these two factors. See supplemental material for details.

**Scoring:**  For each PASCAL $5^i$ fold, 4 mean IoU numbers are computed for each of the QC splits. These are then combined into two scores for the entire dataset - the low-complexity accuracy score (LCA) and the high-complexity accuracy score (HCA). These are computed as weighted averages:

$$LCA = \mu_{folds}\left(\frac{\sum_{i=0}^{i=3} w_i mIoU_i}{\sum_{i=0}^{i=3} w_i}\right), \qquad HCA = \mu_{folds}\left(\frac{\sum_{i=0}^{i=3} w_{3-i} mIoU_i}{\sum_{i=0}^{i=3} w_i}\right) \qquad (2)$$

The $\mu_{folds}$ operation averages the weighted mIoU scores over the PASCAL $5^i$ folds. The two scores are different only in the weights used for averaging the mIoU values for each split. For LCA, the splits with easy or salient query images are given higher weights ($w$=3,0.75,0.75,0.5) and for HCA the weights are reversed to give more importance to the higher complexity images. The weights are proportionate to the simplicity of the query image for each split across the folds. The simplicity factor is computed as the product of the salience of the target object and the IoU with the supervised (DeepLab v3+) network. The factors are also adjusted for computational convenience. More details about the dataset design and statistics are included in the supplemental material.

### 4.1.2 Support Cognizance

As discussed in section 3.3.1, the networks predict some foreground pixels in the masks even when the support information is spurious.   We also observe that the predicted masks are not always of high quality, even when the support image is identical to the query.   Interpolating between these two extreme cases, we construct a set of test pairs with the following levels of similarity between the support and query images:

**L0: Identical**      $I_q = I_s$      Both inputs are identical.
**L1: Transformed**      $I_q = X(I_s)$      $X$ is a random rigid transformation.
**L2: Similar**      $F(I_q) \approx F(I_s)$      $F$ represents the CLIP image features [24] of the image.
**L3: Dissimilar**      $F(I_q) \not\approx F(I_s)$      $I_s$ and $I_q$ are visually dissimilar.
**L4: Diff. Class**      $C(I_q) \neq C(I_s)$      Implies that a different class of object is to be segmented.
**L5: Null Mask**      $M_s$=0      Implies that no target class object is present in the support.
**L6: Empty**      $I_s$=0      A trivial output is expected.

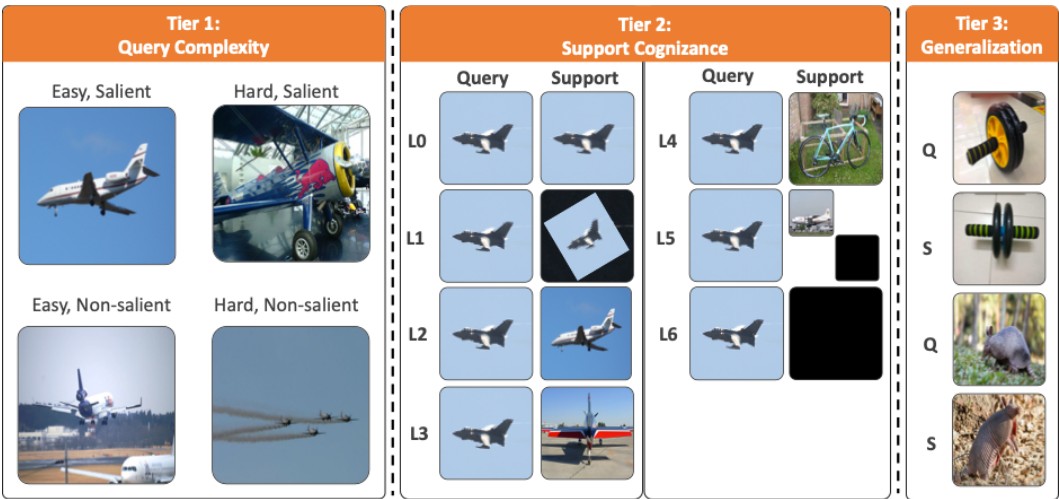

Figure 3: The TOSS Evaluation Dataset. Tier 1 is derived from the PASCAL $5^i$ dataset [26] and consists of 4 splits for which the types of query images are indicated (easy/hard refers to accuracy obtained with DeepLab v3+ [4]). Tier 2 consists of 7 levels of support-query similarity, and Tier 3 has images from the FSS-1000 dataset [13] with no overlap with the PASCAL $5^i$ dataset.

.

For each PASCAL $5^i$ fold, one test file is constructed by sampling an equal number of pairs from each level. A pre-trained CLIP [24] model is used for measuring image similarity because it discerns fine-grained differences between images. It is possible to use other feature extractors for measuring similarity but the results are not likely to be different because the split is made about the mean similarity values for the image pairs (see supplemental material). Note that the accuracy for levels L0-L3 is required to be high for a good solution, and for levels L4-L6, it ought to be lower. To incorporate these considerations, we compute a Support Cognizance score using the mean over all the test classes of the IoU values as follows:

$$SCS^i = Mean_{c \in F^i}(\sum_{j=0}^{j=6} w_j H(j, \frac{\sum_{pairs \in L_j^c} inter(M_q, \tilde{M}_q)}{\sum_{pairs \in L_j^c} union(M_q, \tilde{M}_q)})) \tag{3}$$

$M_q$ and $\tilde{M}_q$ are the ground-truth and predicted query masks respectively. $H(j, x)$ is $(1 - x)$ for L4-L6, and is otherwise the identity function. The standard mIoU computation is modified using the formulation above. First, the IoU is computed for each pair in level $L_j^c$ for each test class $c$ in fold $F^i$. The $H$ function inverts the IoU for the negative levels (L4-L6). This inversion penalizes the score for any spurious foreground pixels predicted for input that should result in a null output. Next, a weighted average of the partition IoU values is computed with weights $w_j$. We use weight values $w_j$ = {4, 2, 2, 1, 1,1,1} to attribute high relative importance to the easy cases. The idea is that L3-L6 can be considered as adversarial input, designed to extricate the limitations of a network, while L0-L2 are for the intended use of the solution. Finally, the aggregated values are averaged over the classes in the fold. The average Support Cognizance Score over the 4 folds indicates the extent to which the network honors the signal from the support input. Next, we discuss the generalization tier.

### 4.1.3  Generalization to Unseen Classes

The generalization tier comprises of a single file of (query, support) pairs taken from the FSS-1000 dataset [13]. This tier measures the ability of the network to generalize to unseen classes. It determines the utility of the network for real-world application to domain-specific, photographic images where the classes are strictly distinct from the training classes. We submit that the regular test folds of the PASCAL $5^i$ are inadequate for measuring generalization, because many test classes have semantically related classes in the training sets. Consider these pairs of training- and test-classes from the different folds of PASCAL $5^i$ - {(dog, cat), (horse, cow), (motorbike, car), (train, bus), (cat, sheep)}. In each of these pairs, the latter class (the test class) often occurs in a similar context as the former (training class), or shares semantic similarities with it. The generalization tier marginalizes these semantic

similarities that skew the accuracy in favor of these classes, by testing on diverse unseen classes like a ringlet-butterfly, a banana, or a coin.

The classes in the tier are chosen to be conceptually disjoint from those in the PASCAL $5^i$ dataset. To achieve this, we remove all classes in FSS-1000 that have at least one semantically similar class in the PASCAL $5^i$ dataset (see supplemental material for a list of the removed classes). This ensures that a network with PASCAL $5^i$ has never seen images like the ones in this tier. The degree of salience of the target class pixels is also balanced by sampling from the most salient and the least salient images equally. The generalization score $GS$ is the mean of mIoU values across the folds. .

| OSS Method | Tier 1 | | Tier 2 | Tier 3 |
|---|---|---|---|---|
| | LCA (wmIoU) | HCA (wmIoU) | SCS (wmIou) | GS (mIoU) |
| Baseline | 58.59 | 38.06 | **68.82** | 82.29 |
| RPMM* | 54.49 | 34.20 | 54.66 | 78.79 |
| PFENet | **59.75** | 38.08 | 61.36 | 83.68 |
| RePRI | 54.14 | 32.52 | 67.81 | 84.75 |
| HSNet | 59.10 | **38.27** | 64.72 | **87.47** |

Table 4: TOSS Evaluation of current Few-shot segmentation methods. It is noteworthy that the leader for different scores are different, thus illustrating the benefit of using TOSS evaluation for comparison. wmIoU refers to weighted mIoU to indicate that the scores are dimensionally identical to mIoU scores.

The generalization scores are high across the methods for this tier, because most images from the FSS-1000 dataset feature a single, salient object.

## 4.2 Results and Insights with TOSS Evaluation

Table 4 presents the scores for multiple networks. Note that we have reported aggregate scores across each tier of the TOSS dataset, because reporting mIoU values for each split separately will result in an unwieldy data table that will be harder to analyze and present. While the metrics are good for comparing solutions, the scores for the data splits themselves can be used without aggregation for better analysis. We derive interesting insights from the aggregate scores:

- **Low-complexity images are segmented accurately:** Even older methods like SimPropNet [7] on which the baseline network is based are able to segment low-complexity images well. Thus, for an application that requires segmentation of images with the target object in salient regions only, one can choose a network with fewer parameters for faster inference.
- **High-complexity images are not segmented accurately:** None of the methods do well enough on the high-complexity query images. Therefore these methods cannot be reliably utilized for segmenting objects in scenarios with a high degree of clutter.
- **mIoU Accuracy versus Precision:** The baseline network is more cognizant of the support image signal than the other networks. This implies that in scenes with objects of multiple classes, where the support may be supplied for one of the many classes, the other networks are more prone to making erroneous pixel classification.
- **Anomalous inversions:** RePRI [3] generalizes on unseen classes better than PFENet [28], but has worse accuracy for the TOSS test set, especially the high-complexity images.

## 5 Experimental Setup

We use the training protocols set for PASCAL $5^i$ as defined in [26]. For one-shot inference, we depart from the prevalent practice of using random query-support pairs (with a fixed seed). For all inference runs, we use a large number of fixed pairs that are shared as a part of the dataset release to ensure fairness of comparison. Also, we include the training set images in the test splits while ensuring that the target class for each such image is disjoint from the training set.

We work with three networks - a baseline network based on SimPropNet [7], the RPMM variant of the Prototype Mixture Models [33], and the Prior-guided Feature Enhancement network [28]. For the latter two, we use publicly available official implementations [34] and [27] respectively. We use

hyper-parameters identical to the ones suggested by each paper. RPMM is trained afresh without filtering out images containing test-class objects to ensure fairness of comparisons. The PASCAL $5^i$ dataset is used for training and testing.

For analysis of the TOSS dataset, we perform inference on publicly available pre-trained models for four methods - RPMM [33], PFENet [28], RePRI [3], HSNet [21], along with the baseline network based on SimPropNet. We use a $U^2$-Net [23] network model pre-trained on the DUTS-TR [31] dataset for salience guidance. Inference analysis is done on single V100/A100 GPUs with 24-core Intel Xeon processors and 160 GB RAM available on the Google Cloud Platform.

**Baseline Network Implementation Details:** The network is trained on PASCAL $5^i$, the input images are of size 512x512 pixels, the learning rate is set at 0.0025, the batch size is 8, and the training is done using stochastic gradient descent with momentum (0.9) for 180 epochs for each fold. The models are trained on Google Cloud virtual machines with four Tesla V100 GPUs and 32-core Intel Xeon processors, and 64 GB RAM.

## 6   Limitations and Conclusions

**Limitations:** We acknowledge that no dataset can capture all the nuances of the OSS problem. We propose this dataset because we find large variance in reported and reproduced results. Even with these nuances considered, not all problems may be discovered. However, the framework of tiers for evaluating different aspects of the problem can be easily extended along other dimensions to ensure consistent reporting.

**General Impact:** We hope that this work will result in more uniform reporting of the accuracy of solutions to the one-shot image segmentation problem. This will improve baseline determination in future research which, in turn, will limit the usage of GPUs for experiments and the associated power consumption.

**Ethical Issues:**   We concede that the dataset does not address any ethical issues present in the underlying datasets. We discuss underlying data biases, as well as issues pertaining to private information here.

*Biases:* The TOSS dataset is a useful reorganization of the existing PASCAL $5^i$ and FSS-1000 datasets with added constraints. It does not introduce any new data biases, or alleviate or exacerbate existing ones. The PASCAL $5^i$ consists primarily of inanimate objects, and the person class, and is devoid of any professional, gender, or racial indications. The subset of the FSS-1000 dataset that we use, considers classes of inanimate objects, animals, and insects.

*Private Information* TOSS is based on the PASCAL $5^i$ dataset which in turn is based on the PASCAL Visual Object Challenge dataset [6]. The images in this dataset are collected from the Flickr photo-sharing website [2] and does contain faces of people. The dataset inherits the Flickr terms of use for images which presents as a deterrent against any untoward intent in using these images. Since setting up the TOSS dataset requires setting up the PASCAL VOC dataset as well, the same terms of use are inherently applicable. The subset of the FSS-1000 dataset that is used by TOSS includes only two images with any discernible human faces. This is determined by running a pre-trained object detection network [16] looking for the person class.

In this work, we present evidence of negative inductive biases in one-shot image segmentation (OSS) methods - namely, the class negative bias and the salience bias - arising from improper treatment of the training data. We also present a new, tiered evaluation dataset for OSS that creates opportunities for a better understanding of the problem and cogent reporting of progress.

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
