# OpenReview forum: "What Ails One-Shot Image Segmentation: A Data Perspective"
_NeurIPS.cc/2021/Track/Datasets_and_Benchmarks/Round2 — NeurIPS 2021 Datasets and Benchmarks Track (Round 2)_

### Official Review · Reviewer_WNFT · 2021-09-18
**Great analysis of two issues, but the proposed benchmark/metrics has several issues**

**Rating:** 6
**Confidence:** 3

**Strengths:**

- The authors identify two key issues with the current common approach to training and evaluating. This would allow others to better understand the failure modes of one-shot segmentation models. The proposed issues also provide insight regarding the current challenges in this domain; eg, handling non-salient or background objects.
- The authors support their observations by a combination of insightful analysis into the problem, clarifying examples, and a series of experiments to validate those hypotheses.
- The proposed benchmark augments existing evaluation datasets in a simple but insightful way. Furthermore, one could imagine applying the same splits to other datasets. Specifically, I liked the Tier 2 benchmark tasks of evaluating the model on its ability to segment the image correctly for both positive and negative queries. Testing a model’s ability to handle a lack of input is very important.
- Departing from the use of random query-support pairs [L275] is a very good idea as it allows for better comparison instead of only relying on the results reported in each paper and assuming prior methods are being represented accurately and fairly.


**Weaknesses:**

[W1] Tier 3 of TOSS claims to test generalization between conceptually disjoint classes, but it’s unclear if it’s accomplishing that.

The paper states that Pascal 5i and FSS-1000 are “conceptually disjoint” [L257] without much justification.This is done in the context of generalizing to unseen classes. I see three issues here.

1. One-shot segmentation evaluates images on unseen classes, so the labeling of the third tier is applicable to the entire problem. Perhaps across-dataset generalization might be a more appropriate label, but it’s unclear why this would be interesting if the datasets are similar (same object classes or same collection method).
2. The class list of FSS and Pascal share a lot of broad categories in common; both datasets have classes that can be classified as vehicles, people, animals, and indoor objects (the classification of Pascal).
3. The performance reported on this tier appears to be higher than performance on Pascal 5i (comparing table 4 with table 1). This is surprising when considering that generalizing to a different domain (L211) usually is more challenging. While it’s possible that FSS is simply an easier dataset resulting in the higher mIoU values, the authors do not discuss that. Finally, there is little motivation or justification as to why generalizing from the classes in Pascal 5i to FSS is a good measure of generalization, especially when considering points 1 and 2.

I would like to emphasize that I think evaluating on conceptually different classes is interesting and important; eg, evaluating on vehicles having been trained only on animals. My critique is that I am not sure if the current split evaluates that any more than splitting the classes in Pascal 5i.

---------------

[W2] The utility of the proposed metrics is unclear.

The authors do a great job at identifying different attributes and dimensions for analyzing OSS performance. Those attributes are used to split the test pairs into different splits. However, instead of simply showing performance along the separate splits, the authors propose a series of aggregated metrics to combine the performance across all splits (Eq 2 and 3). This aggregation includes both weighting the performance on different splits [L234] or flipping IoU metrics [L250]. I think the aggregation is both unnecessary and needlessly confusing, and diminishes from the value that could be provided from analyzing the performance along for the proposed splits.

It is unclear to me why the aggregation is necessary. I found the proposed splits to be very interesting, and I think simply showing the mIoU along each of the splits in tiers 1 and 2 would allow us to analyze the model performance.

Furthermore, I found the aggregation confusing for two reasons. First, the weighting is very arbitrary and not well motivated in the paper. One could imagine that different weightings would favor different method-dataset pairs, making it hard to interpret the results. Second, the aggregation greatly conflates the performance along each of the splits. For example, if we consider the proposed SCS metric (Eq 3), a method could achieve the same SCS value by either doing better on positive support (L0-L3) or doing better on negative support (L4-L6). As a result, the SCS score would conflate those two models by giving them the same score. I think simply reporting the performance along those splits would be more informative.

---------------

[W3] Limited discussion of benchmarked methods

The paper is motivated by the need for a better benchmark to allow for “fine-grained insights” into the performance of the OSS models. While the suggested tiers have the potential to do that, the actual discussion of the benchmarked methods and their relative performance is very limited in Section 4.2. While it is nice to provide high-level findings, I think it is important to both provide more details regarding the methods being compared, as well as more analysis to substantiate the “fine-grained analysis” promised by the authors.


**Additional Feedback:**

Feedback is provided in no particular order.

- Figure 1: I found the first figure slightly confusing. For example, for the Saliency bias portion, it is unclear what the task is. The left position would imply the task is to find the plane, but I am unable to clearly see the image. Furthermore, the class negative bias portion utilizes two colors (orange and white) for masks, but it’s unclear what orange refers to here.

- L70: there seems to be a typo using both determine and gauge in succession.

- Sec 3.2 and Figure 2: It would be good to clarify that DeepLab v3+ is a supervised baseline. While the subplot header indicates that, I think stating it in the caption/text would make it easier to understand the point being made.

- Sec 3.2 and Figure 2: I think the point being made by the authors here is that models with poor OSS test accuracy result from their presence in the training data. However, the plot being shown doesn’t clearly show that. First, it’s unclear which model is being shown in the right portion of Figure 2: relative performance between classes 3,4,5 does not match the OSS or Supervised model from the left pane.  Second, it is unclear if comparing the performance of the model with its presence supports the point being made. Rather, one would be interested in seeing the relative performance drop compared to presence. If so, then a plot showing the correlation between those two variables would be more convincing.

- The examples provided in Figure 3 for L4 and L5 seem flipped.

- Table 2: this table exceeds the right margin of the document. Furthermore, the naming of the rows is very confusing and requires a lot of cross referencing between the text and the figure to understand what is being tested. I would suggest that the authors use more descriptive row headers and reformat the table to improve readability.

- L242: Why is CLIP features a good measure of image similarity with respect to this task? While I could imagine this being a useful heuristic, its use here is not supported by empirical results (experiments or prior work) or some justification in writing.

- L410: broken reference.

- L251: The authors discuss weighting the different elements of the SCS metric using w_j, but it is unclear how w_j is computed.

- The notation for equations 2 and 3 are not standard with some of the elements in equation 3 being undefined (eg, M_q).

- The word “complexity” is used to refer to the first tier of the TOSS, but it’s not clear what the authors actually mean by that. Furthermore, the first tier splits the test pairs according to saliency and difficulty of each pair (evaluated based on a current method). It’s unclear how complexity can refer to those two dimensions.


**Clarity:**

The clarity of the presentation was highly variable within the paper. While I found the introduction and discussion of the biases clear, sections 4 and 5 were less clear to me. I think a few factors contributed to this.

First, the naming of the tiers does not match what they each test (See weakness 1 and additional feedback for a more detailed discussion of this).

Second, several portions of those sections include references to experiments without proper discussion. For example, the attribute choices [L222-230] discuss many factors that the authors considered, however, the long list of factors is not helpful without some explanation as to what they each mean or refer to. I would encourage the authors to either omit those details and simply refer the reader to supplementary or to provide some explanation to how this happened. As it currently stands, this portion takes a lot of space without allowing the reader to understand what the authors actually did.

Third, a lot of design choices are not well substantiated. This applies to the proposed aggregated metrics and their weighting schemes (weakness 1), the use of CLIP to determine image similarity, as well as the choice of methods to compare. I think the paper would be greatly improved if the authors provide some justification or reasoning for those choices.


**Correctness:**

With the exception of claims regarding the conceptual disjointness of classes in TOSS Tier 3 (discussed in the weakness section), the claims in the paper appear to be correct.

**Documentation:**

Github repo seems to provide the required information to run the benchmark.


**Ethics:**

The authors do not discuss ethical implications of their work (L421). There is some discussion of the impact of the paper with respect to improved evaluation of research as well as reduced GPU usage. If the authors believe that this work has no ethical dimension, I would suggest making this explicit.

**Relation To Prior Work:**


The authors contextualize their work with respect to prior work. I noticed one missing citation from a recent conference. Lddecke and Ecker (“The Role of Data for One-Shot Semantic Segmentation”, CVPR-W 2021) discuss some issues in current one-shot image segmentation datasets. While the issues identified by Luddecke and Ecker are different from the ones discussed in this submission, I would encourage the authors to discuss their work.




**Summary And Contributions:**

This paper identifies two biases that affect one-shot image semantic segmentation: class-negative bias and salience bias. In one-shot image segmentation, the model takes as input a support image and an associated mask identifying the object with the task being generating a mask for another query image of the same class as the masked object. The first bias occurs when test classes are dealt with as background during training making it more difficult for the network to identify them at test time. The second bias occurs when the network identifies the salient object in the test image regardless of the query mask. The authors propose some mitigations for those biases as well as a new benchmark to disentangle those issues.

While the analysis is quite interesting and useful, I think the paper has three major weaknesses: (1) the utility of the proposed metrics in TOSS is unclear, (2) it is unclear how tier 3 of TOSS is conceptually disjoint, and (3) limited discussion and analysis of benchmarked methods and performance.

Since some of the weaknesses affect the benchmarking portion of the paper, my rating is a borderline. I discuss those issues in the respective sections of the review, and I would appreciate some clarification and justification from the authors regarding those issues.

--------------
**[Update 9/28]** Raised score to 6/10 in response to author's rebuttal. See comment below for more details.

---

### Official Review · Reviewer_uR6D · 2021-09-19
**Great contribution; missing ethics checklist items**

**Rating:** 7
**Confidence:** 2
**Correctness:** As far as I can tell, the findings an…

**Strengths:**

1. Benchmark exposes new types of bias (bias from the choice of support image & bias from salient non-target objects inthe foreground) that are otherwise difficult to measure.
2. Benchmark effectively addresses specific types of performance bias - well-scoped. Authors demonstrate how the new benchmark provides new insights into current methods (4.2).
3. Supplemental analysis of class negative and salience bias thoroughly motivates the need for a more nuanced benchmark - and proposed mitigation methods show a consistent increase in performance, strengthening this conclusion further.
4. Construction of benchmark maps directly to evidence of bias presented - partitions are based on supervised accuracy and mask overlap.

**Weaknesses:**

1. **[Major]** The authors answered "N/A" for a couple of the ethics considerations in the author checklist, but clearly have not addressed relevant ethical questions relating to their use of PASCAL and FFS. See the ethics section of my review. I think this paper should be accepted, but not published until these checklist items are met.
2. **[Major]** The authors claim at least twice (L11, L195) that their new benchmark is intended to "map better to real-world image segmentation expectations," but it's not clear to me how the benchmark improves over prior work in this aspect. The claimed contribution of added nuance is clearly supported, but this claim isn't, as far as I can tell - no new images are added and no real world applications are specifically examined. Perhaps the authors are arguing that the added focus on complexity and salience maps closer to real world expectations? L195-L200 need to be clarified to make it clear exactly how TOSS maps more closely to real-world expectations. What evidence is there for this claim?
3. **[Minor]** Evidence for class negative bias and salience would be stronger if supplemented with systematic evidence of erroneous masking, as shown in Fig 1. Does the phenomena shown in the Figure happen? Is there a way to describe specifically this phenomenon other than by a general increase in mIoU? The current evidence (especially #3, on L146) seems possibly circumstantial but fairly convincing anyways - a more direct demonstration woluld increase my confidence in the authors' explanation.
4. **[Minor]** Little to no discussion of limitations, especially of the evidence presented for saliency and class negative bias.

**Additional Feedback:**

N/A

**Clarity:**

The writing is very clear. Figures 1 and 3 are great - well done. The only confusion I had was related to Weakness #1.

The clarity of the Github and supplementary material could use some work, though - fortunately, most important information seems to be contained in the paper itself.

**Documentation:**

The documentation is sparse - which may be excused, because reproducibility is fairly simple because the benchmark is based on existing datasets. The codebase contains code for reproducing that seems fairly usable. The attribute choices are based on fixed values (the accuracy of DeepLab, mask overlap, support cognizance score) and not any subjective judgment as far as I can tell. If there was an subjective judgment involved with constructing the splits, that should be included in the documentation to aid replication.

**Ethics:**

The authors don't discuss ethics in the paper or the documentation - before this is published, I think an ethics discussion needs to be added.

The authors' benchmark extends PASCAL and FFS - do these datasets contain images of people? That would constitute personally identifiable information that needs to be mentioned. Looking at the FFS homepage linked on the authors' Github, I immediately see a support image containing a picture of a person with face unblurred. Likewise, the PASCAL paper shows an image of a person on a bike, face unblurred. Figure 1 of this paper includes 3 people! Did PASCAL/FFS ask participants for explicit consent? Were they compensated? Does your use of the dataset promote unethically collected images? Are there any NSFW images in these datasets?

Also, are there any demonstrated instances of bias contained in either of these datasets? For example, COCO object annotations have [demonstrated gender biases](https://arxiv.org/pdf/2004.07999.pdf) (through object scale, co-occurrence, scene diversity, and appearance diversity). This should be discussed as part of the ethics checklist - I don't know of any bias issues with these datasets, but the authors need to check before they re-use those datasets and state in the paper any concerns there might be.

I suspect this benchmark is ethically okay - or at least doesn't compound existing problems in PASCAL and FFS - but the authors need to consider these questions seriously and address it in the supplementary materials - "N/A" does not work here since there are clearly images of people in these datasets. Right now, I don't think this paper meets the guidelines - for example, 4d of the checklist in the paper is certainly applicable and has not been addressed.

I think this paper should be accepted, but not published until these checklist items are actually addressed.

**Relation To Prior Work:**

**Strength**: The authors position their work in relation to PASCAL 5 and COCO 20, two existing benchmarks. Neither benchmark specifies exact pairs of test images, so performance can vary, as the authors demonstrate - this paper addresses that gap and specifically targets salience bias as well.

**Weakness:** The authors should explicitly state whether there is any related work on salience or class negative bias.

**Summary And Contributions:**

This paper proposes a new benchmark dataset for one-shot image segmentation (OSS) intended to close the gap between current evaluation datasets and performance in practice. The authors also present evidence for two types of bias plaguing OSS methods using their benchmark dataset.

---

### Official Review · Reviewer_dspi · 2021-09-20
**Technical Review**

**Rating:** 6
**Confidence:** 4

**Strengths:**

S1 - The paper identifies issues in current OSS methods and introduces a benchmark to test for these issues. This is valuable for the community to further improve existing methods and to be sensitive to the discovered biases.

S2 - The findings are interesting and the evaluation metrics are specifically designed for the discovered issues, creating a good tool for practitioners to investigate the performance of their models.


**Weaknesses:**

W1 - the main weakness of the paper is the clarity of the presentation that makes it difficult to follow the evaluation and insights. See “Clarity”.

W2 - some choices seem to be somewhat arbitrary, such as the weights for combining the different mIoU scores ($4,0.75.0.75,0.5$). Further, I do not understand the choice to invert the mIoU score for L4-L6. Since in all three cases, the GT mask is background only, the intersection is always 0 and the union is equal to the prediction. Only a prediction of background only would result in a non-zero score. This creates a binary metric, that is either 0 or 1 which does not seem to be ideal. Other unmotivated choices: why CLIP features (image+text) to compute image similarities? Does the choice of saliency detector/saliency dataset matter? What are the individual outcomes of the metrics before averaging (this would give much more insights)?

-- post rebuttal --
I have read the other reviews and the authors' responses. The clarity has been sufficiently improved and the paper discusses then inherited ethical concerns of the used datasets. Overall, the presented benchmark is useful to the community and I do not have remaining major concerns to oppose publication.

**Additional Feedback:**

The image quality of the figures is low.

Fig2 and Fig3 have a dot in the center of the page 5 lines below the image.

Table 2 extends beyond the page margins.

Varying number of significant digits in Tab 2.

typos
L147 the gain the accuracy


**Clarity:**

The clarity can be improved. Several notations are not introduced. For example in table 2, the row labels need to be understood by guessing as the notation (e.g. $M_s$ or $C(I_q)$) has not been used before (or after). Also phi is used instead of maybe a zero or an empty set. The $\mu$ function in Eq.2 is not clear to me. The text seems to suggest that this function does the weighted averaging, however the equation applies $\mu$ to the average.

In the same vein, several experiments are not explained thoroughly. The leaveout experiment in table1 only has one line (146) of explanation, the rest needs to be inferred.

The paper refers to the supplementary material in several places for further explanation on the dataset design choices and details. However, the supplement only contains one page with links to the code and external datasets and does not contain the promised explanations. There are some details in the documentation of the github repository (committed after the deadline) but even there the points in W2 are not clarified.


**Correctness:**

The evaluation seems to be mostly correct, a definite answer can be given once the clarity issues have been solved.


**Documentation:**

It is not fully clear whether the paper presents a benchmark or a dataset since it introduces metrics as well as new splits for existing datasets. I would suggest to create a dataset documentation (e.g. datasheet) to discuss the content, copyright and hosting of the used datasets and the new splits.


**Ethics:**

The broader impact section could discuss some of the privacy and bias issues of image datasets that have been scraped from the internet and are being used here.


**Relation To Prior Work:**

Prior work is discussed adequately. The paper makes a strong point about what is missing in previous dataset splits and aims to alleviate these issues.


**Summary And Contributions:**

The paper investigates the task of one shot image segmentation where given a query image and a support (image + mask) as segmentation of the query needs to be predicted. The findings are that current datasets bias the methods towards segmenting salient objects, that background classes in training influence the testing results and that some methods tend to ignore the support input. The paper then introduces dataset splits and evaluation metrics that alleviate these issues and allow for a more fine-grained evaluation of the individual problems.

---

### Decision · Program_Chairs · 2021-10-10

**Decision:**

Accept

**Comment:**

All reviewers recommended accept after considering author responses. AC doesn't find grounds to overturn this consensus.